# A Review of Genetic Polymorphisms and Susceptibilities to Complications after Aneurysmal Subarachnoid Hemorrhage

**DOI:** 10.3390/ijms232315427

**Published:** 2022-12-06

**Authors:** Jose Medina-Suárez, Francisco Rodríguez-Esparragón, Coralia Sosa-Pérez, Sara Cazorla-Rivero, Laura B. Torres-Mata, Aruma Jiménez-O’Shanahan, Bernardino Clavo, Jesús Morera-Molina

**Affiliations:** 1Research Unit, University Hospital of Gran Canaria Dr. Negrín, 35010 Gran Canaria, Spain; 2Department of Specific Teaching Methodologies, University of Las Palmas de Gran Canaria, 35004 Gran Canaria, Spain; 3Instituto Universitario de Enfermedades Tropicales y Salud Pública de Canarias, Universidad de La Laguna, 38296 Tenerife, Spain; 4CIBER de Enfermedades Infecciosas, Instituto de Salud Carlos III, 28029 Madrid, Spain; 5Neurosurgery Unit, University Hospital of Gran Canaria Dr. Negrín, 35010 Gran Canaria, Spain; 6Department of Medical and Surgery Sciences, University of Las Palmas de Gran Canaria, 35016 Gran Canaria, Spain; 7University of La Laguna, 38200 Tenerife, Spain; 8RETIC de Investigación en Servicios de Salud en Enfermedades Crónicas (REDISSEC), 28029 Madrid, Spain; 9Instituto de Salud Carlos III, 28029 Madrid, Spain; 10Chronic Pain Unit, University Hospital of Gran Canaria Dr. Negrín, 35010 Gran Canaria, Spain; 11Radiation Oncology Department, University Hospital of Gran Canaria Dr. Negrín, 35010 Las Palmas de Gran Canaria, Spain

**Keywords:** polymorphism, aneurysmal subarachnoid hemorrhage, vasospasm, delayed cerebral ischemia, neurological outcome

## Abstract

Delayed cerebral ischemia (DCI) and vasospasm are two complications of subarachnoid hemorrhages (SAHs) which entail high risks of morbidity and mortality. However, it is unknown why only some patients who suffer SAHs will experience DCI and vasospasm. The purpose of this review is to describe the main genetic single nucleotide polymorphisms (SNPs) that have demonstrated a relationship with these complications. The SNP of the nitric oxide endothelial synthase (eNOS) has been related to the size and rupture of an aneurysm, as well as to DCI, vasospasm, and poor neurological outcome. The SNPs responsible for the asymmetric dimetilarginine and the high-mobility group box 1 have also been associated with DCI. An association between vasospasm and the SNPs of the eNOS, the haptoglobin, and the endothelin-1 receptor has been found. The SNPs of the angiotensin-converting enzyme have been related to DCI and poor neurological outcome. Studies on the SNPs of the Ryanodine Receptor yielded varying results regarding their association with vasospasm.

## 1. Introduction

Spontaneous subarachnoid hemorrhage (SAH) represents 5% of all brain hemorrhages and has a very high mortality rate (40%) with a higher frequency of aneurysmal SAH (aSAH) [1]. Approximately 70% of patients who suffer from an SAH develop vasospasm and 30–40% present delayed cerebral ischemia (DCI) [2,3,4]. The pathophysiological mechanisms underlying these complications are not well understood, though recent studies point to a multi-factorial process [5,6].

During the first 72 h after suffering an SAH, there exists a possibility of undergoing a global brain lesion or early brain injury (EBI), which can be caused by a disruption in the cerebral blood flow [7]. At this stage, several pathophysiological processes have been observed including microcirculatory dysfunction [8], angiographic vasospasm [9], generalized cortical spreading [10], microthromboembolisms [11], and neuroinflammatory process caused by oxidative stress [12]. Both neurological status based on the World Federation Neurosurgical Scale (WFNS) and blood extravasation estimated by the Fisher CT scale upon hospital admission are considered predictors of neurological outcome [13,14].

The only treatment approved by the Federal Drug Association to prevent vasospasm following an aSAH is nimodipine [4,5,15], although clazosentan, an inhibitor of the endothelin receptor, has proven to be effective in reducing its morbidity [16]. Nimodipine is a calcium channel blocker, and its capacity to cross the blood–brain barrier gives it high selectivity for cerebral vessels. However, not all patients respond to this drug successfully. This variability may be due to different factors such as age, sex, comorbidities, and metabolic variations due to individual genetic susceptibility which is determined, among other factors, by the presence of polymorphic genetic variants [17].

## 2. Genetic Polymorphisms

The aim of this review is to describe the influence of single nucleotide polymorphisms (SNPs) on the susceptibility to complications following an aSAH. We have included several SNPs which have been associated with two of the main complications following an aSAH (vasospasm and DCI) as well as with the neurological outcome (NOC), which can be either favorable or poor (Figure 1).

For each gene presented in this review, we will refer to the SNPs that authors have analyzed to determine if there is any association with these three variables (DCI, vasospasm, and neurological outcome) (Table 1).

### 2.1. Nitric Oxide Synthases

In the brain, neurons and glial and endothelial cells express the enzyme nitric oxide synthase (NOS) which synthesizes nitric oxide (NO) [18]. NOS is considered a regulator of calcium channels, and the NO produced by this enzyme reduces their current in neurons [19]. The lack of regulation of these channels causes neural damage [20]. Therefore, NO synthesized by neural NOS (nNOS) could exert neuroprotective action through these channels [21].

In arterial vessels, NO synthesized by endothelial NOS (eNOS) induces vasodilation through the activation of calcium channels and prevents platelet aggregation [21,22]. This action is very important in the cases of SAH, as thrombi are produced to prevent blood extravasation [8]. Thus, alterations to the production of NO could give rise to pre-thrombotic states, one of the key factors considered in producing DCI [11]. Moreover, polymorphic eNOS may also be responsible for oxidative stress. This may affect blood vessels, thereby participating in the production of vasospasm [23].

eNOS enhances neuroprotection through a process of ischemia/reperfusion, activated after the onset of an SAH [24]. Nevertheless, after 6–72 h, there is an increase in NOS isoform (iNOS) and the basal levels of NO produced by this isoenzyme. iNOS is responsible for some of the neurotoxic effects, which may be related to DCI [25]. In some laboratory studies, DCI improved after the administration of isoflurane as a result of the increase in eNOS expression [26,27]. The importance that NO plays in the effects of SAH has encouraged studies that seek to establish a relationship between the SNPs of *eNOS*, and the origin and complications of aSAHs [23,28,29,30,31,32,33].

In a case–control study including 142 patients, Khurana et al. [23] found that heterozygosity T/C of the SNP *eNOS* T-786-C rs2070744 was associated with larger aneurysms. In a case–control study with 249 patients on the relationship between eNOS and the formation and rupture of aneurysms, Song et al. [34] also studied the presence of the SNP *eNOS* T-786-C rs2070744. In their results, they found that 88.5% of controls were homozygous for the savage T/T allele, whereas 11.5% presented the mutant T/C allele. None of the patients showed the homozygous mutation C/C. There were no significant differences in the distribution of the SNP *eNOS* T-786-C rs2070744 in cases or controls. Therefore, they could not establish a relationship between this SNP and the possibility of suffering an aSAH or DCI. However, they did find that the SNP *eNOS* T-786-C rs2070744 was associated with poor neurological outcome.

There is variability in the clinical manifestation of cerebral vasospasm in patients with an identical score on the Fisher CT scale. In a case–control study with 141 patients, Khurana et al. [28] studied genes codifying proteins which could be aberrant and play an important role in such variability. In this case, they also focused on the *eNOS* and included the SNPs *eNOS* intron 27 VNTR, *eNOS* T-786-C rs2070744, and *eNOS* G-894-T rs1799983.

The authors established the relationship between these SNPs and the score patients had on Fisher CT and WFNS scales upon admission, the presence of vasospasm during hospital stay (be it clinical, radiological, or both), and the dismissal status. The authors found a significant difference in the distribution of the SNP *eNOS* intron 27VNTR in cases compared to controls. There was a predominance of having at least a 4a allele in cases, thereby concluding that carrying an abnormal putative 4a allele (heterozygosity) was related to a risk of up to three times higher of suffering an aSAH. On the other hand, 21 out of the 28 patients that had, upon admission, Fisher CT grade III developed vasospasm. Of these patients, 86% were heterozygotes for the SNP *eNOS* T-786-C rs2070744. All 21 had a significant tendency to carry the C allele, being present in the 57% presenting vasospasm (in 80% of those who presented asymptomatic vasospasm and in 100% of those who presented symptomatic vasospasm).

Ko et al. [35] obtained similar results in their prospective cohort study with 347 patients. This study was carried out to determine the risk of suffering vasospasm. The authors analyzed three SNPs associated with the *eNOS*, registering the presence of angiographic vasospasm (48%) and radiologic infarction (35%). They found a relationship between the SNP *eNOS* T-786-C rs2070744 and cerebral vasospasm, particularly in patients who were homozygous for the C allele (CC).

Starke et al. [36] also studied the relationship between the SNP *eNOS* T-786-C rs2070744 and the presence of cerebral vasospasm. In a prospective cohort study that included 77 patients, they found that 43% presented either symptomatic or angiographic vasospasm, 20% presented infarction secondary to the rupture of aneurysm, and 10% presented infarction secondary to vasospasm. Unlike previous studies, this study showed that the T allele was a predictor of vasospasm. Specifically, they found a significant increase in symptomatic or angiographic vasospasm in patients with TT genotype (61%) versus the CT (35%) or CC (0%) genotypes. The patients that presented the T allele had a 3.3 times higher risk of suffering vasospasm. In the case of the TT genotype, the risk was 10.9 times higher.

Hendrix et al. [29] developed the study CARAS (Cerebral Aneurysm Renin Angiotensin System), a prospective cohort study including 149 patients which was conducted between 2012 and 2015. They studied the relationship between eNOS and DCI through the analysis of the *eNOS* SNP T-786-C rs2070744. Out of all patients, 21.2% presented DCI and 80.6% of these also had clinical vasospasm. An association was found between the C allele of the *eNOS* T-786-C SNP rs2070744 and DCI. However, no association was found between this allele and poor neurological outcome or clinical vasospasm, unlike in other studies [28,35].

During the first 24 h after an SAH, the bioavailability of NO is decreased [37] and its administration has shown positive effects on the microcirculatory spasms which happen at that early stage [38]. However, the endogenous inhibitors of NO, asymmetric dimethylarginine (ADMA) and protein kinase C, increase during this critical 24 h period [37,39]. The increase in ADMA has been associated with DCI, thus making it a biomarker for this complication as well as for poor neurological outcome [40]. This increase can result in dysfunction in the production of the endothelial NO and the regulation of blood flow, leading to vasospasm [40,41].

Hannemann et al. [42] studied the association between the ADMA–NO pathway gene (*DDAH1*) and the risk of suffering DCI after an aSAH. In their cohort study involving 47 patients, they determined serum–ADMA concentrations and the presence of DCI, cerebral infarction, and poor neurological outcome [42]. They found that 18 patients developed DCI and also presented serum–ADMA concentrations significantly higher than the other patients.

They analyzed different SNPs for the *DDAH1* gene, finding significant association between DCI and three SNPs *DDAH1*: rs480414, rs1241321, and rs233112. They also ascertained an association between the SNP *DDAH1* rs233112 and high serum–ADMA concentrations.

### 2.2. Haptoglobin

The hemoglobin released after the rupture of an aneurysm inhibits the production of eNOS, thereby decreasing the concentration of NO in the cells of the smooth muscle and giving rise to vasoconstriction [43]. The spontaneous degradation of oxyhemoglobin to methemoglobin has been related to the release of superoxide, further lipid peroxidation, and vasoconstriction [39]. The products of this degradation buffer the levels of free NO, inhibiting the function of eNOS and causing a serious depletion of NO levels, triggering numerous adverse effects in the cerebral parenchyma [44].

High levels of hemoglobin found after aSAHs have been related to poor neurological outcome [45]. The haptoglobin (HP), a protein involved in the protection against oxidative damage, attaches to the hemoglobin and prevents its degradation and the consequent oxidative stress which may affect neurons [43].

In an attempt to find the association between both eNOS and *HP* SNPs and aSAH complications, Lai and Du [30] conducted a meta-analysis in which they studied the relationship of these to DCI and radiological and angiographic vasospasm. This meta-analysis included 20 studies with a total number of 1670 patients. The authors analyzed 27 SNPs in 11 genes.

In their conclusions, Lai and Du observed that the SNP *eNOS* VNTR a allele was associated with DCI in opposition to the b allele. They also found a relationship between the *HP* 2-2 allele of the HP and radiographic vasospasm. They analyzed the SNP *eNOS* T786C rs2070744 as well but did not discover any association with poor neurological outcome [30].

Morton et al. (2020) [46] conducted a study with a cohort of 907 patients in which they found association between the *HP* 2-2 allele and a favorable neurological outcome. This association was detectable two years after patients suffered an aSAH. It was present in patients who had suffered a high-volume aSAH but was not present in those who had suffered a low-volume aSAH.

### 2.3. Endothelin-1

In an aSAH, there is an imbalance between the vasodilator effects of NO and the vasoconstrictor effects mediated by the endothelin which cause oxidative stress and inflammation [39]. In fact, endothelin-1 (END1) plays an important role in vasospasm. Some clinical trials showed an improvement in vasospasm, but not neurological outcome, when the endothelin receptor antagonist clazosetan was used [47].

From the data collected through the CARAS study, Foreman et al. [31] observed that 3.4% of patients had presented rebleeding, which is related to poor neurological outcome. Following this finding, these authors analyzed the association between rebleeding and the *END1* SNPs. Their results showed that the T allele for the SNP *EDN1* G/T rs2070699 was an independent factor for rebleeding in these patients.

Additionally, using the data from the CARAS study, Grissenauer et al. [48] analyzed the association between the END1 and clinical vasospasm, DCI, and poor neurological outcome. The END1 functions by binding to two receptors: ENDRA and ENDRB. ENDRA (coded by the *ENDRA* gene, located in chromosome 4) can be found mainly in smooth muscle cells in cerebral vessels. Grissenauer et al. analyzed the SNPs of the genes responsible for the codification of both ENDRA and ENDRB and END1. In their results, they found that the T allele of the SNP *END1* T/G rs1800541 is associated with the presence of an aSAH, whereas the G allele of the SNP *ENDRA* G/C rs5335 is related to clinical vasospasm.

### 2.4. High-Mobility Group Box 1

After suffering an aSAH, there are several prothrombotic processes which can cause cerebral infarctions in different brain areas [49] where arteriolar constriction and low NO levels have been found [44]. This stimulates vasodilation, allowing the arrival and activation of leukocytes and inducing the synthesis of iNOS which triggers and increases the inflammatory response through the release of cytokines. Moreover, microglia and astrocytes also express iNOS in proinflammatory states, producing superoxides [25,50]. This excitotoxicity produced after the infarction is accompanied by an increase in neuronal calcium influx [51]. The high level of intracellular calcium could deactivate the modulation of GMPc through NO in patients who suffer from DCI [47]. These changes cause an expansion of the depolarization which produces neural electric dysfunction, arteriolar vasoconstriction, and neurovascular uncoupling [52]. This inflammatory state can lead to the production of reactive nitrogen species which can damage the DNA [53], activating or deactivating transcriptional factors [54,55] and provoking alterations in the physiological metabolism and cell apoptosis [56,57].

High-mobility group box 1 (HMGB1) is a ubiquitous protein involved in the promotion of inflammatory response in the presence of cell damage. High levels of this protein have been related to poor neurological outcome and a high risk of suffering cerebral vasospasm in patients who have also suffered an aSAH [58]. A study developed by Hemmer et al. [59] showed that high levels of HMGB1 upon hospital admission predict the appearance of DCI.

With the data obtained through the CARAS study, Hendrix et al. [60] analyzed the association between the SNP *HMGB1* rs2249825 and the presence of DCI (identified in 21.2% of the patients included in the study). They found that the G allele for this SNP was associated with a high risk of suffering DCI after an aSAH.

### 2.5. Serpin Family E Member 1

Hendrix et al. [60] also studied the association between prothrombotic states post-aSAH and the Serpin Family E Member 1 (*SERPINE1*) SNPs. *SERPINE1* gene codes the plasminogen activator inhibitor-1 (PAI-1), which is a precursor of plasmin, the predominant proteolytic enzyme in fibrinolysis. By inhibiting the activation of the plasminogen, the PAI–1 blocks fibrinolysis.

Their results showed that no *SERPINE1* SNP was associated with clinical vasospasm. Nonetheless, patients with AA and 4G/4G genotypes of the SNP *SERPINE1* rs2227631 showed a higher risk of developing DCI. Moreover, patients with the GG genotype of the SNP *SERPINE1* rs7242 and the AA genotype of the SNP *SERPINE1* rs2227684 showed favorable neurological outcome.

Lin et al. [33] conducted a case–control study with patients from the study CARAS. They also analyzed the *SERPINE1* gene and its relationship with the risk of suffering SAH and presenting a poor neurological outcome. They studied six SNPs: rs2227631, rs1799889, rs6092, rs6090, rs2227684, and rs7242. Their results showed that the G allele of the SNP *SERPINE1* rs2227631 was significant for suffering an aSAH in the case group. In this group, they also observed that the recessive homozygous for G5GGGT was associated with DCI, clinical vasospasm, and a long ICU stay. Additionally, the carriers of G5GGAG were associated with lower incidences of cerebral edema and a higher Glasgow scale score, whereas the carriers of A4GGGT were associated with lower incidence of severe high blood pressure.

### 2.6. Renin–Angiotensin–Aldosterone System

The Renin–Angiotensin–Aldosterone System (RAAS) plays a fundamental role in the process of vasodilation/vasoconstriction of cerebral vessels as well as in the vascular remodeling and maintenance of the arterial wall’s integrity. The walls of aneurysms show less presence of the angiotensin converting enzyme (ACE) and angiotensin II receptor type 1 (AT1), which causes a thinning of the arterial wall and a lack of vascular remodeling [61].

Through the data obtained in the CARAS study, Grissenauer et al. [32] analyzed the relationship between vasospasm, DCI, and neurological outcome and the following *RAAS* SNPs: *AGT* C/T rs699, *ACE* I/D rs4340, *AT1* A/C rs5186, *AT2* A/C rs11091046, and *AT2* G/A rs1403543.

The results showed that the I allele of the SNP *ACE* I/D rs4340 was associated with poor neurological outcome [32]. Moreover, in patients ≤55 years old, there was an association between DCI and SNP *ACE* I/D rs4340, whereas in patients >55 years old, the association was between DCI and the A allele of the SNP *AT2* A/C rs11091046.

### 2.7. Ryanodine Receptor

The Ryanodine Receptor (RYR) activates calcium channels, which are the main target of the nimodipine, by regulating the diameter of the smooth muscle cells and the cerebrovascular tone.

Rueffert et al. [62] conducted a cohort study with 46 patients who had suffered an aSAH. They analyzed the association between the SNPs *RYR* rs34934920 and *RYR* rs35364374 and cerebral vasospasm, finding that heterozygous G/T carriers of the SNP *RYR* rs35364374 had significantly higher incidence of symptomatic vasospasm compared to carriers of the G/G genotype.

However, using the data from the CARAS study, Hendrix et al. [63] also studied the SNP *RYR* rs35364374 and its association with vasospasm, DCI, and poor neurological outcome and did not find any association.

## 3. Conclusions

Complications following an aSAH can be related to the existence of genetic variations. Both DCI and vasospasm as well as neurological outcome have been associated with different SNPs. However, the studies centered on these associations so far have focused on the search for specific candidate genes. Carrying out this review, we have not found any genetic study that does not start from a prior proposal. Conducting a genome-wide association study could reveal genetic variations that play a relevant role in the occurrence of complications post-aSAH that were previously unknown.

## Figures and Tables

**Figure 1 ijms-23-15427-f001:**
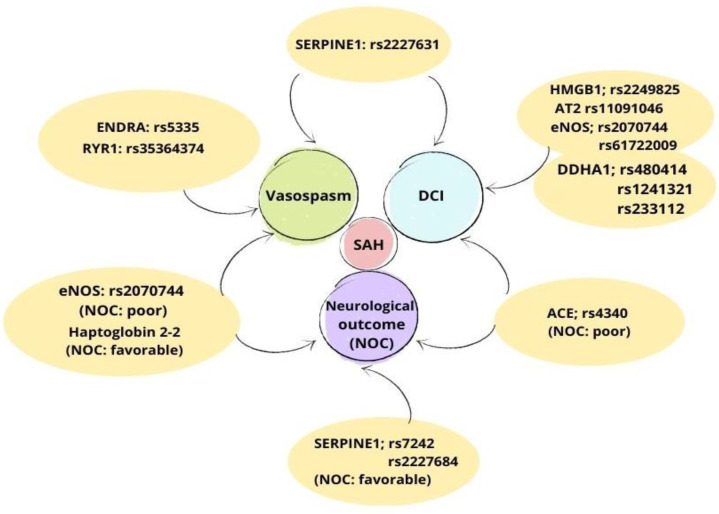
Associations of neurological outcome (NOC), delayed cerebral ischemia (DCI), and vasospasm with single nucleotide polymorphisms (SNPs).

**Table 1 ijms-23-15427-t001:** Genes and single nucleotide polymorphisms (SNPs) and their associations with neurological outcome (NOC), delayed cerebral ischemia (CI), and vasospasm.

Gene	SNP	NOC	DCI	Vasospasm
*eNOS*	rs2070744	A (poor)	A	A
	rs61722009	NA	A	NA
*DDAH1*	rs480414	NA	A	NA
	rs1241321	NA	A	NA
	rs233112	NA	A	NA
*HP2-2*	-	A (favorable)	NA	A
*ENDRA*	rs5335	NA	NA	A
*HMGB1*	rs2249825	NA	A	NA
*SERPINE1*	rs2227631	NA	A	A
*AT2*	rs11092046	NA	A	NA
*ACE*	rs4340	A (poor)	A	NA
*RYR*	rs35364374	NA	NA	A

A: association was found; NA: no association was found.

## Data Availability

All data are present in the manuscript.

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
