# Peer review of "A Review of Genetic Polymorphisms and Susceptibilities to Complications after Aneurysmal Subarachnoid Hemorrhage"

_ijms, 2022, doi:10.3390/ijms232315427_

Round 1
Reviewer 1 Report
The information summarized and discussed in this review will be beneficial for early stage drug discovery research. a few important questions for the authors to consider-
1) What was the rationale for choosing the genes HMGB1,eNOS and others?
2) How did authors identify these genes, when there are several other genes whose SNPs have been implicated in SAH and related complications. For example, ApoE mutations have been implicated for enhanced risk of SAH in Asians; Ace, Col4a, Methylenetetrahydrofolate Reductase (Mthfr) and several others in Caucasian and Asian population.
3) It will be beneficial to expand the list of genes and shed light on the clinical trial landscape for these targets for SAH or associated indications
4) here are examples of some references to help the authors to conduct a deep dive on their ideas- https://www.ahajournals.org/doi/10.1161/STROKEAHA.117.017072
https://www.ncbi.nlm.nih.gov/pmc/articles/PMC6321144/
https://jamanetwork.com/journals/jamanetworkopen/fullarticle/2782340
https://www.ncbi.nlm.nih.gov/pmc/articles/PMC3716358/
https://www.ahajournals.org/doi/10.1161/STROKEAHA.120.032621
https://www.ahajournals.org/doi/10.1161/str.52.suppl_1.P42
5)Why is the manuscript related to genomics of SAH (Neuro/Cerebrovascular disease) submitted under the special issue of "Molecular Genetics of Cancer" ?
Author Response
Dear sir/madam,
Thank you very much for reviewing our work and for the comments and suggestions you give to us. They have been very useful to us.
We would like to proceed to answer your questions:
1) The reason why we have chosen the genes included in the manuscript was because after doing some research we found that these are the ones that have shown higher association with the complications after SAH, as well as with the outcome of patients who have suffered it.
2) Regarding other genes such as ApoE, Mthfr or ACE I/D, it is true that we have found their association related to intermediate phenotypes. However, we finally decided to include those for which we had found a more direct and frequent association with the complications of the SAH.
3 & 4) We would like to thank you for giving us some references to conduct a deeper analysis of other genes and their relationship with cerebral hemorrhages.
The studies conducted about intracerebral/intraparenchimal hemorrhage and genetic variants (Falcone & Woo; Chen et al; Jha et al) are studies we will definitely consider for future studies in which we widen the scope of hemorrhagic events further than SAH.
We have found extremely interesting the work done by Weinsheimer et al regarding genetic variants and vasospasm, as it is closely related to the work we have revised. We will definitely follow the future studies these authors publish on this topic and will take them into account in our future work.
5) Regarding the submission to the special issue "Molecular Genetics of Cancer", we have sent an email to the academic editor apologizing for what may have been our mistake in the process of submission, as it was not our intention to send it to be published in that issue. We have also suggested them other issues for their consideration of the publication.
Finally, we have sent the manuscript to a proofreader to double check the text for inconsistencies or mistakes we have may made regarding the use of style of English language.
Thank you very much for your time and invaluable contributions.
Yours sincerely,
the group of authors.
Reviewer 2 Report
The authors presented a review article regarding the influence of the single nucleotide polymorphisms on the susceptibilities to complications after aneurysmal subarachnoid hemorrhage. They have included several single nucleotide polymorphisms which have been associated with the main complications after aneurysmal subarachnoid hemorrhage - vasospasm and delayed cerebral ischemia. The article is well written and concise. It tells us where we stand and in which direction we should focus on in the future regarding genetic polymorphisms and susceptibilities to complications after aneurysmal subarachnoid hemorrhage.
Author Response
Dear sir/madam,
We would very much like to appreciate your reading of our manuscript and thank you for your words and the consideration given to our work. It is very significant and important to us that you consider that the review settles the moment we are at and the direction towards which the studies in this topic head.
Yours sincerely,
the group of authors.
Round 2
Reviewer 1 Report
Good job in revising the manuscript and expanding the gene sets to a larger pool that are relevant to human pathology.